# Automatic Video-Oculography System for Detection of Minimal Hepatic Encephalopathy Using Machine Learning Tools

**DOI:** 10.3390/s23198073

**Published:** 2023-09-25

**Authors:** Alberto Calvo Córdoba, Cecilia E. García Cena, Carmina Montoliu

**Affiliations:** 1Escuela Técnica Superior de Ingenieros Industriales, Center for Automation and Robotics, UPM-CSIC, Universidad Politécnica de Madrid, José Gutiérrez Abascal St., 2, 28006 Madrid, Spain; 2Escuela Técnica Superior de Ingeniería y Diseño Industrial, Center for Automation and Robotics, UPM-CSIC, Universidad Politécnica de Madrid, Ronda de Valencia, 3, 28012 Madrid, Spain; cecilia.garcia@upm.es; 3Instituto de Investigación Sanitaria-INCLIVA, 46010 Valencia, Spain; carmina.montoliu@uv.es; 4Servicio de Medicina Digestiva, Hospital Clínico de Valencia, 46010 Valencia, Spain

**Keywords:** machine learning, brain functionality, diagnosis, medical applications, automatic video-oculography system

## Abstract

This article presents an automatic gaze-tracker system to assist in the detection of minimal hepatic encephalopathy by analyzing eye movements with machine learning tools. To record eye movements, we used video-oculography technology and developed automatic feature-extraction software as well as a machine learning algorithm to assist clinicians in the diagnosis. In order to validate the procedure, we selected a sample (n=47) of cirrhotic patients. Approximately half of them were diagnosed with minimal hepatic encephalopathy (MHE), a common neurological impairment in patients with liver disease. By using the actual gold standard, the Psychometric Hepatic Encephalopathy Score battery, PHES, patients were classified into two groups: cirrhotic patients with MHE and those without MHE. Eye movement tests were carried out on all participants. Using classical statistical concepts, we analyzed the significance of 150 eye movement features, and the most relevant (*p*-values ≤ 0.05) were selected for training machine learning algorithms. To summarize, while the PHES battery is a time-consuming exploration (between 25–40 min per patient), requiring expert training and not amenable to longitudinal analysis, the automatic video oculography is a simple test that takes between 7 and 10 min per patient and has a sensitivity and a specificity of 93%.

## 1. Introduction

Data analysis in the medical context has been usually based on classic statistics, but the emergence of high-throughput technologies and the need to incorporate them into decision making has provoked the incorporation of modern machine learning approaches. In fact, among the different domains in which machine learning (ML) is developed, healthcare has been pointed out as one of them due to its great impact on society and its possible development and application, as has been widely proven [1].

Nowadays, ML is starting to be applied in clinical practice for diagnosis. Using previous diagnosis, it is possible to train ML algorithms in order to improve diagnostic performance. Moreover, by applying ML to recorded data, it is also possible to predict the evolution of a disease or to prevent complications. ML algorithms have dealt with the scarcity of specialized doctors, the rejection of non-specialized doctors, and an increased use of unnecessary or inappropriate tests. Machine learning allows the generation of expert systems that, based on objective quantification of neurological tests from clinical examination, can improve the diagnostic capacity of non-specialized doctors and even support expert doctors in decision making. One of the most complex parts of the neurological examination is the evaluation of eye movement disorders. Routine clinical examination is insensitive to subtle abnormalities, and sometimes, the use of instrumental systems is required. Even with these systems, the interpretation of eye movement remains complex and requires significant expertise. Solutions to simplify the evaluation of eye movement disorders are highly desirable, and ML could potentially contribute to a better diagnostic and prognostic decision making.

Video-oculography is a well-known technique to analyze alterations in eye movements. There are three main methods to measure eye movements:1.Scleral Search Coil (SSC). The transducer element is a contact lens with a coil that is physically attached to the pupil. Although it is considered the gold standard of measurement, the system is an invasive method that is uncomfortable for patients, and it must be handled by an expert [2].2.Electro-oculography (EOG). The transducer element is a group of electrodes attached around the eyeball. The electrical potential generated by the muscles is recorded with electromyography devices. This technique was the most used until the last decade, even though noise signals are common [3].3.Video-oculography (VOG). Here, one or more cameras record the eye movements, and gaze data are computed thanks to the processing of the images. Nowadays, this non-invasive technique is the most used [4,5], but this method needs to be standardized.

Due to technological advances, the measurements of eye movements and the objective quantification of some potential alterations arise as a rich source of information to improve diagnosis [6,7,8,9,10,11,12,13]. In spite of this, the available technology is not suitable for clinical practice because the results are difficult to understand, requiring time-consuming interventions of scarce experts as they are not related directly to a particular behavior of the brain disease. This limits the use of eye movement in research environments and decreases its implementation in routine clinical practice.

Recently, more evidence has arisen in the literature related to the sensitivity and sensibility of eye movement to assist in cognitive impairment by measuring eye movements [14,15,16,17]. Additionally, authors have experience in the development of devices for measuring eye movement [18,19], as well as in feature extraction and analysis in people affected by a neurological disease [20,21,22,23].

Minimal hepatic encephalopathy (MHE) is the earliest form of hepatic encephalopathy and can affect up to 80% of cirrhotic patients. By definition, it has no obvious clinical manifestation and is characterized by neurocognitive impairment in attention, vigilance, and integrative function [24]. The gold standard for the diagnosis of MHE is a time-consuming psychometric test called the Psychometric Hepatic Encephalopathy Score (PHES) [25,26] that must be corrected by the age and education level of the patient, and then highly specialized personnel are required to perform the PHES battery. As a consequence, MHE remains under-diagnosed because the current gold standard is not a reliable procedure to be used in clinical practice. Moreover, some studies reveal that the PHES battery is not sensitive enough to assess an early diagnosis [27,28,29], while eye movements can significantly improve the diagnosis [30].

The PHES is a battery of five psychometric tests—the digit symbol test (DST), the number connection test A (NCT-A), the number connection test B (NCT-B), the serial dotting test (SD), and the line tracing test (LTT)—which evaluates mainly mental processing speed, motor speed, attention, and visuo-spatial coordination. In spite of this consensus, the PHES battery has some drawbacks. In the last decade, different groups have realized that the PHES is not sensitive enough to detect all patients with mild cognitive and/or motor impairment. There is a population of cirrhotic patients who are classified as *not having MHE* by the PHES but show impaired performance in psychometric tests assessing certain neurological functions more specifically [27,28,29].

Therefore, there is a clear necessity to find new tools and procedures for the early detection of MHE in a feasible and reliable way in clinical settings.

New approaches include different physiological signal-processing methods, including cardiac and cerebral activity, thanks to the analysis of electrocardiogram (ECG) and electroencephalogram (EEG) signals. While [31] proposed a normalization and optimization of PHES thanks to EEG analysis, ref. [32] analyzed different parameters extracted from a P300 wave, obtaining statistically significant differences. Other approaches are focused on cardiac activity: Reference [33] built a deep learning model for the diagnosis of cirrhosis. Moreover, the diagnosis of hepatic encephalopathy in cirrhotic patients is also tackled with ML models trained with unbalanced data [34]. Most of these features are general demographic data.

This paper presents a novel approach to assist in the diagnosis of MHE in a clinical setting based on high-resolution eye movement measurements with VOG technology and machine learning techniques. The result is a reliable, non-invasive, and affordable medical tool to be applied in clinical settings.

The concept is sketched in Figure 1. After recording specific eye movements with an IR camera at 100 FPS, computer vision algorithms are used to obtain the position, velocity, and acceleration of the patient’s gaze. After computing the main features of the recorded signals, the most significant of them (*p*-value ≤ 0.05) are used to train ML algorithms. Finally, a PDF file report is sent back to the clinician with measured values and ML results.

In order to validate our proposal, we applied it to a group of patients with liver cirrhosis. Forty-seven cirrhotic patients were included, twenty-four of whom had been diagnosed with minimal hepatic encephalopathy (MHE). This particular impairment is common in 30 to 50% of cirrhotic patients.

This article is organized as follows: Section 2 describes the material and methods used to achieve a suitable algorithm. Section 3 shows the main results. Finally, the discussion and conclusion of the results are detailed in depth.

## 2. Materials and Methods

### 2.1. Material

#### 2.1.1. Video-Oculography System

The gaze tracker used to measure eye movement is based on the VOG technique to record eye movement [18]. An infrared camera at 100 FPS is used to capture the movement of the dominant eye. A conventional chin rest is used to prevent head movements; then, a pure eye movement is measured. The chin rest is placed at 60 cm from the screen used to deploy visual stimulus. This is a typical setup in video-oculography systems [19,22,23,35,36]. The hardware consists of three key elements:

Structural components, which are the body of the device. They are in charge of the correct fitting of the camera through two degrees of freedom.Fixation components, which prevent head movements during tests. Patients may place their heads on the forehead and chin rest.Recording components, which include the camera, infrared LEDs, hot mirror, and both monitors. To ensure eye recording and the use of infrared light while patients are undergoing tests, a hot mirror allows the reflection of IR light and data recording, while patients can follow stimuli on the monitor through it thanks to its characteristic transparency.

Table 1 shows the technical characteristics of the measurement.

The Human–Machine Interface (HMI) is a relevant element of this system. This platform has two HMIs deployed on two screens. One of them is used to show visual stimuli according to medical protocol. This screen is 60 cm from the capture system. To ensure this monitor has a higher refreshing rate than the human sampling rate (60–80 Hz), the refreshing rate of the stimuli in the monitor is 120 Hz.

The other screen is for medical use. Here, the technician can configure the video-oculography session and visualize the result of measurement. Moreover, the data acquisition process, video processing for pupil detection, and a “demo” version of each test are some of the main functionalities of this interface.

#### 2.1.2. Patients

The current gold standard for MHE diagnosis is a battery of psychometric tests called PHES (Psychometric Hepatic Encephalopathy Score) [25,26]. PHES comprises a battery of 5 psychometric tests: digit symbol test (DST), number connection test A (NCT-A), number connection test B (NCT-B), serial dotting test (SD), and the line tracing test (LTT) [25]. DST test evaluates processing speed and working memory, NCT-A and NCT-B evaluate mental processing speed and attention, and SD and LTT are related to visuospatial coordination.

The global PHES scores were calculated with Spanish normality tables (http://www.redeh.org/phesapp/datos.html (accessed on 20 September 2023)) adjusted for age and education level. Patients were defined as having an MHE with a score≤−4 points [25].

This battery is time-consuming, requires expert personnel, and needs manual correction according to age and education [37].

In order to demonstrate how artificial intelligence could be a powerful tool to assist in the diagnosis of a particular disease, we selected a group of patients with liver cirrhosis. Between 30 and 70% of cirrhosis patients suffer a subtle cognitive defect known as minimal hepatic encephalopathy (MHE). This impairment brings a poor quality of life, poor driving performance, and increased mortality. Nevertheless, there are efficient treatments that can reverse symptoms. For these reasons, an early diagnosis is critical for this group of patients [38].

Forty-seven age-matched cirrhotic patients were selected. PHES battery and ocular movement tests were performed on the group. Based on the PHES battery results (Table 2), patients were classified into two groups: twenty-three cirrhotic patients without MHE and twenty-four cirrhotic patients with MHE.

### 2.2. Methods

In this section, we present a brief description related to the steps followed to analyze the data. Details are explained in depth in Section 2.2.2.

Figure 2 shows the procedure to find alterations of ocular movements in cirrhotic patients. The first step is to record ocular movements according to medical protocol; then, OSCANN’s software provides a wide list of features that should be carefully analyzed. The reader is referred to [35] for a full description of the internal process in OSCANN’s software. In order to obtain those typical alterations of eye movement in cirrhotic patients, we performed two different analyses in the Matlab environment (see Figure 2). Initially, we separated variables into parametric and non-parametric using the Shapiro–Wilk test (S-W). If the feature is parametric, *p*-value is computed through the ANOVA test and through Kruskal–Wallis in the other case.

If the *p*-value < 0.01, then the features are considered with potential significance to be included in the machine learning algorithm. In other cases, the feature is discarded from the analysis.

After computing the *p*-value of each feature, we analyzed the accuracy of a group of machine learning algorithms. If it was higher than 85%, we checked the ROC curve, more specifically, the area under the curve (AUC). That area represents a ratio between the true and false values. Finally, we selected the best AUC value and, consequently, the algorithm used to obtain it.

#### 2.2.1. Data Acquisition and Preprocessing

All included patients performed the full battery of ocular movement tests selected by protocol [35]. This protocol was designed to characterize the ocular movements of voluntary participants with no previous cognitive impairment in order to establish a control group. In this case, the biomedical procedure established the following steps:Revision of the patient’s condition.1.PHES evaluation and clinical diagnosis.2.Participants are asked to remove make-up and any kind of lenses before the ocular movement tests in order to guarantee the precision and accuracy of the eye-tracking algorithm.Ocular movement tests: Each test must be clearly explained to the patient, and there is a demo version available if needed.1.Visually guided saccades tests (VGST).2.Antisaccades tests (AST).3.Memory-guided saccades tests (MGST).4.Smooth pursuit tests (SPT).5.Fixation test (FIXT).

Each participant in the experiment followed this protocol based on the analysis and design of a group of specialists that tried to guarantee the attention of the participant. A simple test, VGST, is used to present the experiment. Then, the most difficult tests are captured (AST and MGST), and, finally, the easiest ones are presented to complete the ocular movement experiment.

In order to obtain an accurate ocular movement measurement, the patient’s head must remain fixed; a conventional chin rest is used. The system offers twenty-three different ocular movement tests, but according to the study design following the previous literature and feasibility criteria, cirrhotic patients carried out horizontal and vertical saccadic paradigms (visually guided saccades, memory-guided saccades, and antisaccades) without a gap or overlap and a horizontal and vertical smooth pursuit test. Furthermore, all patients performed a fixation test.

The visual stimulus is a green dot (diameter = 1 cm) deployed on a black background. In each position, the stimulus remains for 1500 ms in each position.

As mentioned previously, the saccadic paradigm includes three tests and each one is performed in the horizontal and vertical directions. Here, eye movements are guided by one stimulus, which appears in the centre of the screen and then moves randomly to the left or right position (horizontal test) or the up or down position (vertical test).

In the visually guided saccades test (VGST), the instruction to the patient is to look at the green dot. The visual stimulus performance is shown in Figure 3.

In the antisaccades test (AST), the patient is asked to look in the opposite direction to the stimulus, such as a mirror. Here, the instruction to the patient is to look to the opposite side of the green dot. Figure 4 shows the concept.

The memory-guided saccades test (MGST) is the longest test in the saccadic paradigms. Here, the stimulus appears in a particular position, remains there for 1500 ms, and then comes back to the screen’s center and disappears. The patient must remember the stimulus position and then perform a saccadic movement toward that. The user has an extra 1500 ms; for this reason, this test takes double the time (see Figure 5).

In this test, the instruction to the patient is to remember the position of the last stimulus and move your eyes toward it.

Table 3 describes the recommended parameters. The visually guided saccades and antisaccade take 36 s and 24 s in the horizontal and vertical directions, respectively, while the memory-guided saccades test takes double the time.

The *Smooth Pursuit* test (SPT) is also performed in the horizontal and vertical directions, and the stimulus moves following a linear wave. The recommended parameters are summarized in Table 4. In linear smooth pursuit, the explored visual field is $\pm 20^{\circ }$, while velocity remains constant and each lap takes 8 s (see Figure 6).

Finally, in the *fixation test* (FIXT), the stimulus remains in the screen’s center for five seconds. The objective of this test is to measure involuntary eye movements such as microsaccades, drifts or square ware jerks, and distractions.

After data acquisition is completed, videos have to be pre-processed in order to compute the position, velocity, and acceleration of the subject’s pupil. Consequently, the patient’s gaze is determined and, and test performance can be analyzed properly.

Before explaining the next steps in our algorithm, it is necessary to focus on the way cirrhotic patients perform eye movement tests. This circumstance is identified due to the inability to obtain data from all tests. Specifically, this problem shows up with bad eye movement recording due to eye morphology; for example, fallen eyelid impairments cause the impossibility of detecting a patient’s line of sight correctly.

Another common situation is patients who could not collaborate in performing all tests because of fatigue, stress, or nervousness. However, this is not the case for cirrhotic patients, since the patients collaborated and the environment where the tests were performed was quiet and comfortable. Then, all patients completed the eye movement tests.

In Section 3, there are more details on the number of patients who have missing tests.

#### 2.2.2. Data Analysis

Before the classification task can be carried out, data analysis process is needed in two distinct phases. On the one hand, feature extraction from ocular movement registration is the first step. On the other hand, the most significant features have to be selected in order to train machine learning algorithms. Therefore, finding the best combination of features allows the best possible classification results to be obtained.

Prioritizing this objective, two theoretical methods are proposed: while signal theory is used for computing features, statistical theory is used for selecting significant features.


**Feature extraction process**


By analyzing the signal generated as a response to the visual stimulus, it is possible to define the features of the eye movement [9]. From each visual test, different features are computed, and then more than 150 features are evaluated.

In VGST, features such as response time or latency toward the stimulus and latency back toward the center of the screen, mean velocity and velocity peaks, accuracy (dysmetria), and the account of the number of blinks or anticipated saccades (the ones too fast made before 80 milliseconds after the stimulus changes) are examples of variables that are measured [39].

In MGST, the account of the number of correct memory saccades, together with visual saccades features, are evaluated in this test. Features such as latencies, accuracy, or velocity features are assessed on the memory saccades.

In AST, when the patient looks in the opposite direction to the stimulus directly, these saccadic movements are considered to constitute correct antisaccade performance. If the patient performs a saccade movement to look at the stimulus and then to look in the opposite direction, these saccade movements are considered “reflexive” saccades, but if they do not look in the opposite direction, there will be an incorrect antisaccade [39]. In this test, other features are measured like in the two previous cases.

In FIXT, as mentioned before, the account of saccades, microsaccades, drifts, square ware jerks, and distractions are measured. Also, different characteristics of each type of micromovement are computed [7,40].

In SPT, it is important to care about the account of catch-up saccades, which are performed by the patient to achieve the stimulus when the gaze is lagging behind it, and back-up saccades, which are performed in the opposite direction of the movement of the stimulus when the gaze is ahead [6]. In addition, some indices related to the time following the stimulus or errors of performance are measured [41].

Finally, Table 5 summarizes the number of features extracted from each test used in this study.


**Significance analysis**


After processing feature extraction, the significance of each ocular movement feature is assessed using classical statistics. First, the normality of each feature must be tested (see Figure 2) to properly generate *p*-values. According to the number of samples (n=47), the Shapiro–Wilk (SW) test is one of the most suitable methods to evaluate the features’ normality [42,43]. Regardless of this, in [44], it is also stated that the SW test can be used with sample sets with more than hundreds or thousands of samples. A second validation of the features’ normality was made using Lilliefors (LF) in order to guarantee feature significance with both methods.

The second step is to obtain the measure of the significance of each feature. To accomplish this, the *p*-value of each variable is computed for assessing the rejection of the null hypothesis. After features were classified as parametric or non-parametric variables, the *p*-values were computed, and just those variables with *p*-values less than or equal to 0.001, 0.005, and 0.01 were included in the following steps.

A one-way analysis of variance (ANOVA) test was used for parametric features in a lot of fields of medicine such as cancer [45] or mammogram mass classification [46], biological data analysis [47], etc. As another parametric test, ANOVA starts from the assumption that the data set fits normally distributed data. This test is a simple case of the linear regression model.For nonparametric variables, the Kruskal–Wallis (KW) test is used [48]. This test is equivalent to ANOVA. In KW, it is not assumed that data obey a particular distribution, and then the normality of data set is not supposed. Furthermore, in [49], it is demonstrated that KW is suitable for a sample set size like the one used in this article.

#### 2.2.3. Classification Algorithm and Validation

The MATLAB™application *Classification Learner* provides a useful tool for training multiple classification algorithms, including parameter variations for a specific algorithm. Therefore, it is a powerful tool to easily test all these algorithms in order to select them properly.

For instance, in the case of the well-known *Supported Vector Machine* algorithm (SVM), linear, quadratic, cubic, and Gaussian kernel functions are available for testing. The SVM classification algorithm has demonstrated very useful qualities such as speed, efficiency, and robustness, which are extremely important for classification tasks [50,51,52].

The best classifiers were selected depending on the accuracy, area under the ROC curve (AUC) [53], and the number of features used. After that, cross-validation was performed using five folds or subsets. One of these folds, which corresponds to 20% of the available samples, was used for testing, while the remaining four were used for training. The samples of each class were equally divided into the different folds.

In this study, we used the cross-validation method where the data set is aleatory, divided into five subsets following a stratified k-fold division strategy. In order to compute the significance and statistical accuracy of the procedure, the selected classifier algorithm was executed 1000 times in an iterative loop, which corresponds with the two last stages shown in the flowchart of Figure 2. All result metrics are computed as the mean value of the errors in each iteration; Figure 7a,b also shows the distribution over these iterations.

Therefore, the training data set and test data set are statistically evaluated. Moreover, the probability of belonging to one of both classes is calculated via predicted class scores [54]. This metric allows us to know if a sample is in the borderline between sets or, even worse, if it is classified in the wrong set.

## 3. Results

The main results of this work are related to the computation of classification algorithms to train and validate SVM classifiers. A few samples were excluded because some of them had missing values in some tests; a full description is included at the end of this section. Section 4 provides details about how to deal with this typical situation in patients with cognitive impairment.

The Classification Learner App provides very useful information about the best algorithm to make the classification task. Table 6 summarizes the results of the best algorithms.

First, the SVM algorithm behaved better than the others, so different kernel functions were computed to know the best way to perform the classification task. Figure 7a,b show the results of this classification study; statistical values of accuracy and AUC are shown as the results of 1000 iterations.

As observed in Figure 7a,b, linear SVM has the best results due to its statistical values in accuracy and AUC. In addition, for a larger detail of the results, scores were computed as the distance to decision boundaries. After that, the probabilities of belonging to a group were compared. Table 7 indicates the score results.

The sigmoid function used to make the transformation of scores to posterior probabilities is represented in Figure 8a, where the slope parameter was −2.826 and the intercept parameter was 0.065.

According to the classification results, the data engineering process was successful thanks to the large amount of significant ocular features. By applying the methodology described above (Section 2.2), Table 8 summarizes the *p*-value threshold computed for each feature.

Then, by considering the *p*-values threshold, there were fourteen features with significant values to discriminate between two groups. These features belonged mainly to memory-guided saccades, antisaccades, and linear smooth pursuit tests, while features from the visually guided saccades test and fixation test are less significant than the first ones. Table 9 presents the measured statistical values expressed as mean value ± standard deviation (SD) for both patient groups and their associated *p*-values.

Finally, missing values in cirrhotic patients appeared as one feature to consider. Table 10 shows the number of patients in each group of cirrhotic patients that have missing values. Section 4 discusses the convenience of being able to include these patients in the classification task.

## 4. Discussion

As established in the introduction section, this study attempts to address a problem concerning the clinical diagnosis of MHE in cirrhotic patients. Currently, this reversible manifestation is under-diagnosed due to the low specificity and sensitivity of its gold standard, the PHES. Furthermore, it is a time-consuming test with subjective nuances based on the age and education of the patient, which requires specialized personnel and makes its clinical practice difficult. In this study, we have focused on the use of eye-tracking technology for the evaluation of eye movement and some cognitive functions to detect this manifestation with better results than the gold standard.

According to the performance of the different classifiers, the linear SVM classification algorithm obtains the best results, as shown in Table 6. Furthermore, KNN and discriminant algorithms achieve at least 90% of maximum accuracy, so everything leads us to remark that the two different classes are clearly differentiated. Figure 8b illustrates how patients are discriminated against using the best two features indicated in Table 9. In this figure, colour represents each cirrhotic patient class while crosses mean misclassification. At this point, it is interesting to point out the great results obtained with linear SVM. This algorithm achieves, as is shown in Figure 7a, 0.935 of mean accuracy, which agrees with two misclassifications, one in each class. In addition, 0.959 mean AUC (Figure 7b) is remarkable, and even coarse Gaussian SVM improved it. This result remains an excellent performance, which ensures data separability.

In the same way, scores and posterior probability results point to visible distances between groups. However, even though twenty-eight out of thirty patients were classified properly, four of them show up within the limits of thresholds, as can be seen in Figure 8a. Numerical results appear in Table 7. Cases with a distance shorter than 0.5 are not considered completely differentiated, which allows them to be called atypical. We do not consider that the detection of complications should be closed categories. There are patients who are borderline that need to be detected and specifically managed. These cases have to be studied and will have more relevance when the number of patients in each group increases. Moreover, Figure 8b shows remarkable differences between classification errors in both classes. While the misclassified cirrhotic patient without MHE is within the limit of thresholds concerning atypical cases, the misclassified MHE patient shows explicit features typical of cirrhotic patients who do not have neurological complications.

According to the statistical correlation between both types of cirrhotic patients, the most significant tests (MGST, AST, and SPT) are the ones that assess cognitive functions such as working memory, inhibitory control, and the capacity to continuously focus their gaze to trace the stimulus position across the visual field. This evidence provides proof that this novel tool is able to detect the manifestation of cognitive impairment in MHE patients according to the previously reported symptoms [28,55,56,57,58]. Moreover, test duration has been reduced to at least a third compared to the current gold standard, which makes it available for better longitudinal analysis.

Concerning the reason why missing values appear in some patients, missing data could arise for many reasons:The patient feels uncomfortable with the device.The patient’s eye movement does not present some particular feature. For example, if there is no reflexive saccade in the antisaccade test, there would be an empty value for this feature.From a medical point of view, the presence of empty values in some measurements could indicate that some particular tasks would present greater difficulty. This difficulty could be associated with the presence of cognitive impairment to some degree. The fact that there are also empty values in the group of patients without MHE could be explained by the fact that there are patients who, although in the PHES and classified as without MHE, may present some cognitive alterations detectable with other more specific tests [28] that the eye-movement test would also detect.

Taking into account the previous comment, there are several reasons for empty values in the data set. Then, in this work, we studied the performance of different algorithms under this condition without considering the potential reasons for them. After looking for the best combination of patients, significant features, and non-missing values, the results show that the neurological evaluation through eye tracking is a promising tool to assist with the clinical diagnosis.

To conclude, the current methodology ensures complete repeatability of the results, as it includes the cross-validation technique. This stage was included due to the biggest limitation of this study, the number of data samples. From the beginning of this study, gathering patients with these cirrhotic manifestations has been the main issue as medical specialists should adapt to the use of the device, patients should be interested in the development of the eye tracking tasks, and the biomedical protocol should be followed to guarantee the highest quality of the samples. To achieve this, training sessions dedicated to each specialist were accomplished, as well as a continuous improvement in the protocols involved.

## 5. Conclusions

In this article, the first automatic video-oculography system to assist in the diagnosis of cognitive impairment in cirrhotic patients is presented. A group of patients with liver cirrhosis was selected. Around 50% of them were diagnosed with cognitive impairment using the medical gold standard, the PHES battery.

Cirrhotic patients carried out the ocular test with OSCANN desk 100. According to the visual tests performed, a set of features was computed and statistically evaluated. The most significant features were extracted from the memory-guided saccades test, the smooth pursuit test, and the antisaccades test. These features were used to train the ML classifier. The algorithm obtained has a sensitivity of 93% and a 93% specificity, which is better than the actual gold standard.

Taking into account the results and the limitations of our research, we plan to increase the sample size of both sets (cirrhotic patients with and without MHE diagnosis) in order to validate the aid to the diagnosis for the clinicians. The sample size of this work is limited but promising, improving the actual gold standard results. Moreover, the high decrease in test duration, its objectivity, and its simplicity make this inexpensive tool especially indicated for improving the help to the clinical diagnosis of this cognitive impairment.

Finally, it would be interesting to analyze the eye movement of patients with MHE under medical treatment to evaluate the effect of the drugs as well as the dose, in improving the MHE of cirrhotic patients.

## Figures and Tables

**Figure 1 sensors-23-08073-f001:**
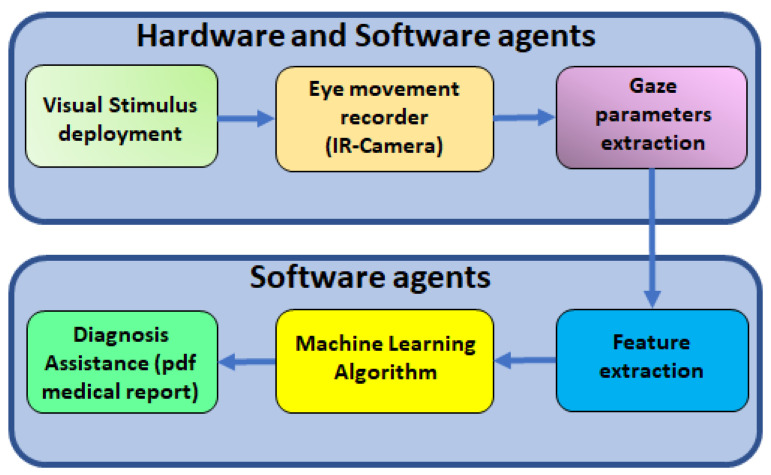
Scheme of the ML concept applied to the diagnosis of cognitive impairment using an automatic video-oculography register.

**Figure 2 sensors-23-08073-f002:**
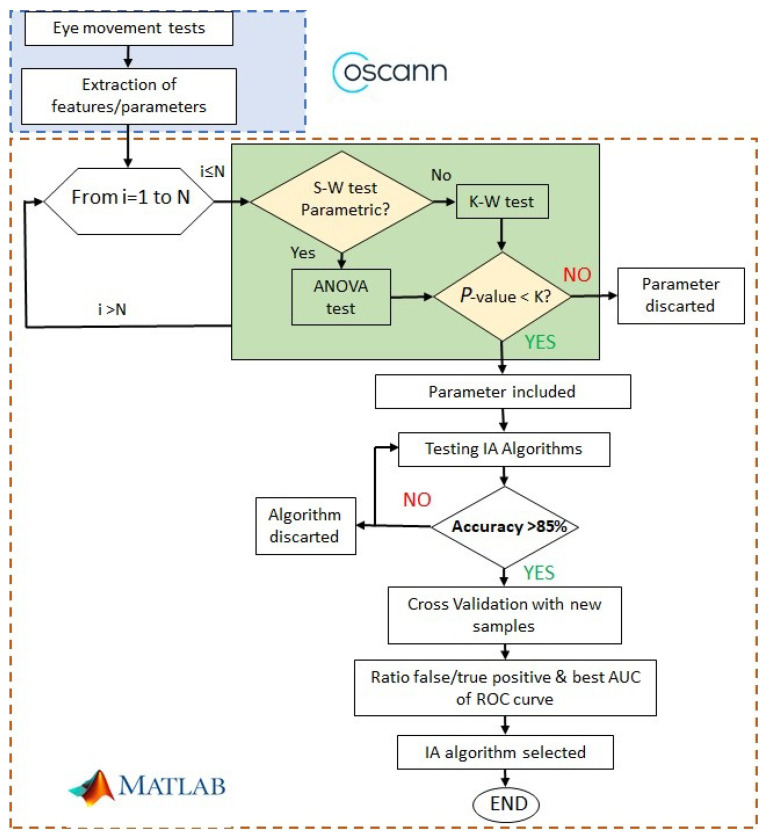
Process followed to select significant features and machine learning algorithms.

**Figure 3 sensors-23-08073-f003:**
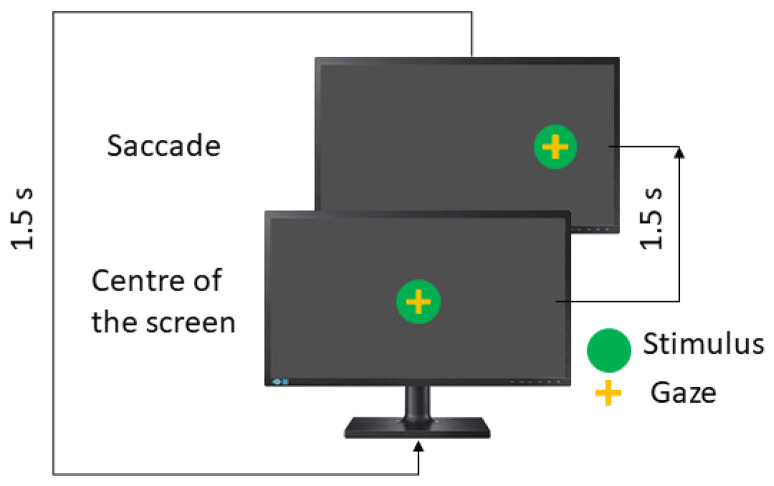
Visual saccades test.

**Figure 4 sensors-23-08073-f004:**
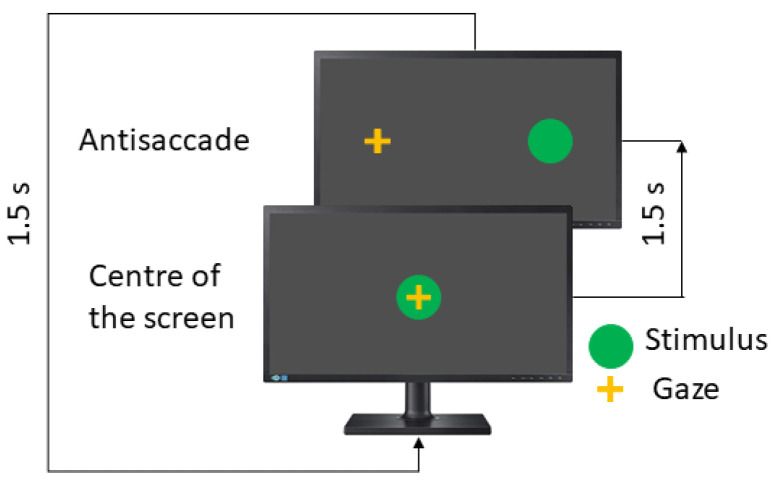
Visual antisaccades test.

**Figure 5 sensors-23-08073-f005:**
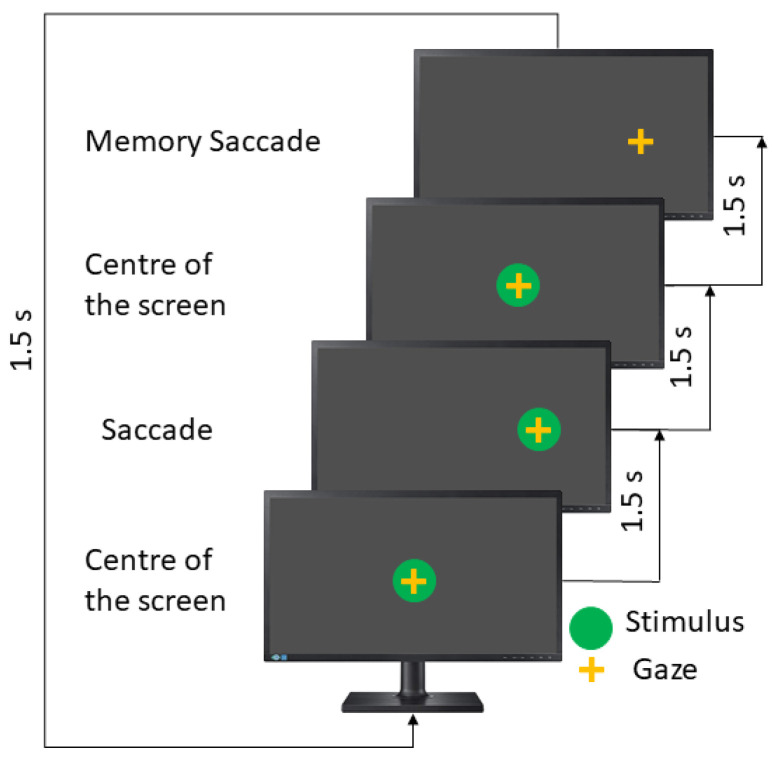
Memory saccades test.

**Figure 6 sensors-23-08073-f006:**
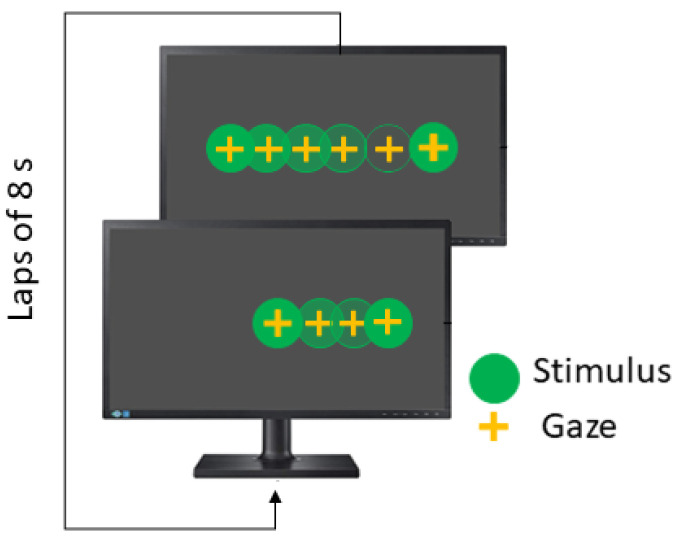
Horizontal smooth pursuit test.

**Figure 7 sensors-23-08073-f007:**
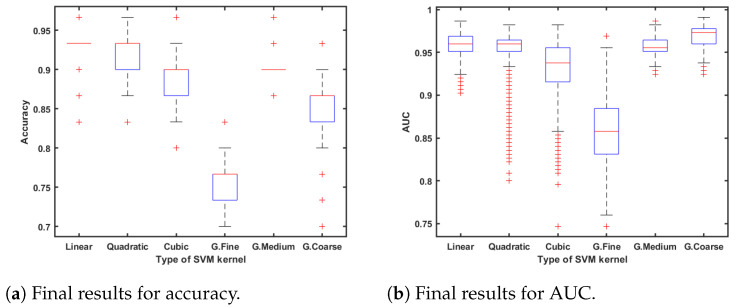
Statistical classification results.

**Figure 8 sensors-23-08073-f008:**
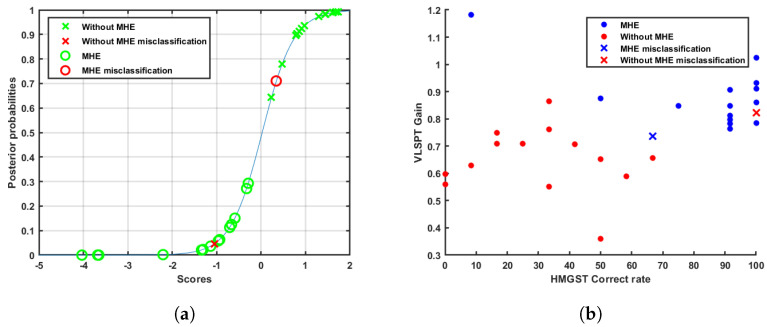
Examples of sample separation. (**a**) Representation of sigmoid function and patients. (**b**) Scatter plot. Patient-discrimination example.

**Table 1 sensors-23-08073-t001:** Gaze-tracker’s technical properties.

Property	Values
Precision	$0.03^{\circ }$
Accuracy	$0.4^{\circ }$
Resolution	$0.0033^{\circ }$
Time resolution	10 ms
Measurement limits	Horizontal: $40^{\circ }$
	Vertical: $24^{\circ }$

**Table 2 sensors-23-08073-t002:** Characteristics of patients in the study and classification according to PHES score. Values are expressed as mean ± SEM.

Patient Type	Without MHE	With MHE
Patients (male/female)	23(19/4)	24(20/4)
Age	60.7±1.6	61±1.8
Educative Level (years)	12±3	12±3
PHES: average scores	−0.75±0.23	−7.26±0.74

**Table 3 sensors-23-08073-t003:** Default parameter for Saccadic Paradigm.

Axes	Visual Field (°)	Duration (s)	Repetitions
Horizontal	5, 10, 20	36, 72	22
Vertical	5, 12	24, 48	12

**Table 4 sensors-23-08073-t004:** Default parameters for Smooth Pursuit test.

Axes	Visual field (°)	Frec (Hz)	Repetitions
Horizontal	20	0.125	3
Vertical	12	0.125	3

**Table 5 sensors-23-08073-t005:** Number of features extracted in each test.

Test Name	Features per Axis
Visual Saccades	18
Memory Saccades	6
Visual Antisaccades	24
Fixation	12
Linear Smooth Pursuit	20

**Table 6 sensors-23-08073-t006:** Best accuracy results for each classification algorithm.

Algorithm	Accuracy (%)
Supported Vector Machine	96.7
K Nearest Neighbors	93.3
Linear discriminant	90
Ensemble: subspace discriminant	90
Ensemble: bagged trees	80

**Table 7 sensors-23-08073-t007:** Mean scores and posterior probabilities after 1000 iterations.

Decision Boundaries Distances D		Sample		Posterior Probabilities	
Without MHE	With MHE	Without MHE	With MHE	Without MHE	With MHE
|D| >1		7	7	*p* > 0.97	*p* < 0.04
1 < |D| < 0.5		5	5	*p* > 0.90	*p* < 0.15
0.5 < |D| < 0.25		1	2	*p* = 0.78	*p* < 0.29
0.25 < |D| < 0		1	0	*p* = 0.65	–
D = −1.05	D = 0.34	1 ms	1 ms	*p* = 0.05	*p* = 0.71

**Table 8 sensors-23-08073-t008:** Significance features of each test.

Test	*p*-Value < $10^{-4}$	*p*-Value < $10^{-3}$	*p*-Value < $10^{-2}$	*p*-Value > $10^{-2}$
Visually guided saccades	0	0	1	17
Memory-guided saccades	2	2	0	2
Antisaccades	0	0	4	20
Fixation	0	0	0	12
Linear smooth pursuit	1	0	4	15

**Table 9 sensors-23-08073-t009:** Values of significance of ocular movements.

Eye Movement Features	Cirrhosis without MHE	Cirrhosis with MHE	*p*-Value
Horizontal memory guided saccades test—correct	10±3.02	4.27±3.23	<$10^{-4}$
Horizontal memory guided saccades test—correct rate	83.33±25.20	35.56±27.0	<$10^{-4}$
Vertical linear smooth pursuit test—gain	0.87±0.11	0.66±0.12	<$10^{-4}$
Vertical memory guided saccades test—correct	6.93±1.71	3.20±3.08	<$10^{-3}$
Vertical memory guided saccades test—correct rate	86.67±21.37	40±38.44	<$10^{-3}$
Horizontal linear smooth pursuit test—pursuit time	92.18±4.39	86.39±6.81	<$10^{-2}$
Horizontal antisaccades test—errors	0.27±0.59	3.47±4.36	<$10^{-2}$
Horizontal antisaccades test—errors rate	2.22±4.95	28.89±36.31	<$10^{-2}$
Vertical antisaccades test—errors	0.2±0.56	2±2.14	<$10^{-2}$
Vertical antisaccades test—errors rate	2.5±7.01	25±26.73	<$10^{-2}$
Horizontal linear smooth pursuit test—gain	0.85±0.12	0.69±0.21	<$10^{-2}$
Vertical linear smooth pursuit test—pursuit and saccades error	1.79±1.22	3.87±3.51	<$10^{-2}$
Vertical linear smooth pursuit test—pursuit error	1.73±1.18	3.70±3.22	<$10^{-2}$
Horizontal visually guided saccades test—latency std	31.92±10.15	55.47±29.56	<$10^{-2}$

**Table 10 sensors-23-08073-t010:** Data set. Selection of the subset with full eye movement parameters.

Groups	Sample Size	Sample without Missing Data
Patients with MHE	24	15
Patients without MHE	23	15

## Data Availability

Restrictions apply to the availability of these data. Data are unavailable due to privacy restrictions.

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
