# Peer review of "Automatic Video-Oculography System for Detection of Minimal Hepatic Encephalopathy Using Machine Learning Tools"

_sensors, 2023, doi:10.3390/s23198073_

Round 1

Reviewer 1 Report

This article introduces an automatic gaze-tracking system for detecting minimal hepatic encephalopathy using eye movement analysis and machine learning techniques. Video-oculography technology was utilized to record eye movements, and an automatic feature extraction software along with a machine learning algorithm were developed to aid in diagnosis. The study involved 47 cirrhotic patients, half of whom had minimal hepatic encephalopathy. Eye movement features were analyzed, with significant ones identified through classical statistical methods and used to train machine learning algorithms. The proposed video-oculography test demonstrated high sensitivity and specificity (93%), offering a faster and simpler alternative to the time-consuming Psychometric Hepatic Encephalopathy Score battery.

The article has major flaws: 

i) Motivation and literature review sections are weak

ii) novelty is not clarified

iii) ML part is very weak for a sensor paper

Overall the paper's methodology is very weak and results are not well formulated. The article needs major re-writing to improve these sections to become standard for Sensors. 

Reviewer 2 Report

Reviewer understand that Córdoba et al. have submitted a manuscript entitled "Automatic videoculography system for detection of Minimal Hepatic Encephalopathy using machine learning tools". Reviewer has a few suggestions and questions, and they would like to request authors to kindly answer all of the questions.

1) In line number 77, "Since the 11th ..." is there. Kindly write "th" in the superscript.

2) In line number 221, "In section??..." is there. Kindly write Section number. Remove all typos from your manuscript.

3) In line number 386, "patients with leaver..." is there. Kindly write correct spelling as liver.

4) Kindly provide Figure number 7,8, 10 and 11 in the HD and bold format. Currently they are blurry.

5)  Kindly discuss about the repeatability of the results using the machine learning tools.

6) Please mention your study's limiting or impacting factors/ parameters at previous stages and how you worked on them in your presented studies.

7) In section 5: Conclusions, please mention about current limitations of your work and the future scope of improvements.

Reviewer 3 Report

General comment:

The manuscript introduces a system for detecting minimal hepatic encephalopathy (MHE) using video-oculography and machine learning (ML) tools. The work is relevant in the field of machine learning-aided sensors for biomedical applications. Furthermore, the proposal is well-motivated and represents an advance in its area. The experimental framework is clear and well-supported. The results are interesting and promissory for detecting MHE with ML techniques. I have some points that should be addressed before the paper can be accepted for its publication.

Comment 1:

The title shows the word “videoculography”; however, it should be “video-oculography” as stated in the abstract. Please, correct it.

Comment 2:

In the abstract, the word “Conclusion” in line 11 seems weird to appear there.

Comment 3:

Some additional references to recent trends in signal processing for detecting MHE should be included. For instance, see

 https://link.springer.com/chapter/10.1007/978-981-99-1916-1_20

https://doi.org/10.14309/ajg.0000000000001617

https://doi.org/10.3390/app13053036

Comment 4:

Some references to figures and sections are missing within the manuscript. They appear as “??” symbol.

Comment 5:

The authors should carefully revise for grammatical errors and typos. E.g. page 5 line, should be “the features are”, instead of “the features is”.

Please, carefully search for grammatical errors and typos.

Round 2

Reviewer 1 Report

Thanks for the work to improve the article. However, the article is still very weak in discussing relevant literature, explaining research design, particularly the results are not presented in the standard format of the sensor journal. This article is missing information on data preparation for ML (i.e., cross-validation), this is a major flaw. Experimental setup should be mentioned clearly as well. Moreover, how this study allows to draw a scientific conclusion is not clear. There are lot of necessary and necessary plots in the paper without proper story which generate little interest among the audience to read the paper. Less important table and image should be moved to supplementary materials. Authors should re-structure the article to make it suitable for Sensor journal and highlight novelty of their work in respect to literature clearly in introduction and discussion section. Mention limitation of their work as well.
